# Decreased Protein Kinase C-β Type II Associated with the Prominent Endotoxin Exhaustion in the Macrophage of FcGRIIb−/− Lupus Prone Mice is Revealed by Phosphoproteomic Analysis

**DOI:** 10.3390/ijms20061354

**Published:** 2019-03-18

**Authors:** Thunnicha Ondee, Thiranut Jaroonwitchawan, Trairak Pisitkun, Joseph Gillen, Aleksandra Nita-Lazar, Asada Leelahavanichkul, Poorichaya Somparn

**Affiliations:** 1Medical Sciences Program, Faculty of Medicine, Chulalongkorn University, Bangkok 10330, Thailand; thunnichaon@yahoo.com; 2Center of Excellence in Immunology and Immune-mediated Diseases, Department of Microbiology, Chulalongkorn University, Bangkok 10330, Thailand; thiranut.rama@gmail.com; 3Center of Excellence in Systems Biology, Research affairs, Faculty of Medicine, Chulalongkorn University, Bangkok 10330, Thailand; trairak@gmail.com; 4Laboratory of Immune System Biology, National Institute of Allergy and Infectious Diseases, National Institutes of Health, Bethesda, MD 20892-1892, USA; joseph.gillen@nih.gov (J.G.); nitalazarau@niaid.nih.gov (A.N.-L.)

**Keywords:** intestinal candida, dextran sulfate solution induced colitis, dysbiosis, bacteremia, gut leakage

## Abstract

Dysfunction of FcGRIIb, the only inhibitory receptor of the FcGR family, is commonly found in the Asian population and is possibly responsible for the extreme endotoxin exhaustion in lupus. Here, the mechanisms of prominent endotoxin (LPS) tolerance in FcGRIIb−/− mice were explored on bone marrow-derived macrophages using phosphoproteomic analysis. As such, LPS tolerance decreased several phosphoproteins in the FcGRIIb−/− macrophage, including protein kinase C-β type II (PRKCB), which was associated with phagocytosis function. Overexpression of PRKCB attenuated LPS tolerance in RAW264.7 cells, supporting the role of this gene in LPS tolerance. In parallel, LPS tolerance in macrophages and in mice was attenuated by phorbol 12-myristate 13-acetate (PMA) administration. This treatment induced several protein kinase C families, including PRKCB. However, PMA attenuated the severity of mice with cecal ligation and puncture on LPS tolerance preconditioning in FcGRIIb−/− but not in wild-type cells. The significant reduction of PRKCB in the FcGRIIb−/− macrophage over wild-type cell possibly induced the more severe LPS-exhaustion and increased the infection susceptibility in FcGRIIb−/− mice. PMA induced PRKCB, improved LPS-tolerance, and attenuated sepsis severity, predominantly in FcGRIIb−/− mice. PRKCB enhancement might be a promising strategy to improve macrophage functions in lupus patients with LPS-tolerance from chronic infection.

## 1. Introduction

The Fc gamma receptor (FcGR) is a receptor for the Fc portion of immunoglobulin and FcGRIIb is the only inhibitory receptor within the FcGR family [1]. The defect of FcGRIIb has been identified as one of the genetic causes of systemic lupus erythematosus (SLE) and FcGRIIb−/− mice, with the loss of the inhibitory signaling, have been used as one of the representative lupus mouse models. Indeed, FcGRIIb dysfunction polymorphisms are common in Asia [2], at least in part, due to the genetic pressure from malarial infection in the region [3]. Although the control of microorganisms in FcGRIIb−/− mice, including malaria, is better than wild-type cells due to the hyperactive inflammatory response [3], these mice are more susceptible to sepsis with its repeated antigenic stimulation because of the severe systemic inflammatory response syndrome [4]. Indeed, the immune exhaustion after repeated- or high-stimulation by pathogens in chronic infection or sepsis, respectively, is well-known [5,6]. LPS tolerance is a helpful mechanism against LPS intoxication, with a decreased response to LPS exposure (mainly through reduced cytokine production) increasing mouse survival in an LPS injection model; in sepsis, LPS tolerance increases patient susceptibility to opportunistic infections and inhibits monocytes from the adequate production of inflammatory cytokines [7,8,9]. These effects suggest a possible link between LPS tolerance and the immune exhaustion phase of sepsis.

We have previously reported the overwhelming response to endotoxin (LPS) followed by the prominent immune exhaustion in FcGRIIb−/− mice [10]. Similarly, patients with lupus are susceptible to infection and this susceptibility towards infection is increased in patients with immune exhaustion [11,12,13,14]. Hence, immune exhaustion is, at least in part, one of the mechanisms leading to the increased infection susceptibility in lupus. Indeed, the mortality rate from infection of lupus patients was increased even prior to the immune-suppression era [11]. In addition, we have reported that the insufficient inflammatory response in FcGRIIb−/− mice is due, in part, to the more prominent endotoxin (LPS) tolerance of FcGRIIb−/− macrophages, despite macrophage hyperresponsiveness after the single dose of endotoxin stimulation [10]. A better understanding of the mechanisms underlying immune exhaustion in FcGRIIb−/− mice might be useful for the clinical management of infection in patients with lupus. 

Because (i) FcGRIIb−/− macrophages (in comparison with wild-type cells) are hyperresponsive following the first dose of LPS activation, but are markedly exhausted after the second dose, and (ii) the post-translational modification by phosphorylation is a common mechanism of rapid protein alteration and functional changes, we postulated that the phosphoproteomes of wild-type and FcGRIIb−/− cells will respond differently to LPS stimulation and performed phosphoproteomic analysis to test this hypothesis.

## 2. Results

### 2.1. Prominent Responses with Profound Exhaustion after LPS Stimulations in the FcGRIIb−/− Macrophage

To demonstrate the different responses to LPS of FcGRIIb−/− and wild-type cells, supernatant cytokine concentrations after LPS stimulation were measured. Increased cytokine production after a single LPS stimulation (N/100) with obvious LPS-tolerance (cytokine-level difference between N/100 and 100/100) of FcGRIIb−/− macrophages over wild-type cells was demonstrated by all cytokines (Figure 1A–C), highlighting the immune inhibition defect of FcGRIIb−/− [4]. In parallel, increased phagocytosis with limited microbicidal activity (due to excessive phagocytosis) of N/100 FcGRIIb−/− macrophages compared with N/100 wild-type cells was evident (Figure 1D–F). In addition, LPS tolerant (100/100) FcGRIIb−/− cells exhibited less phagocytic activity (1h post incubation) than LPS-tolerant wild-type cells and N/100 FcGRIIb−/− cells (Figure 1E,F). Moreover, the microbicidal activity of LPS-tolerant FcGRIIb−/− cells was lower than that of N/100 FcGRIIb−/− cells (Figure 1D), despite the smaller number of phagocytosed bacteria in LPS-tolerant cells (Figure 1E). Hence, the LPS tolerance affected several macrophage functions, including cytokine production, phagocytosis, and microbicidal activity, predominantly in FcGRIIb−/− cells over the wild-type cells.

### 2.2. Significantly Decreased Protein Kinase C-β Type II (PRKCB) in FcGRIIb−/− Macrophage with LPS Tolerance Was Revealed by the Phosphoproteomic Analysis 

To explore the different protein responses of FcGRIIb−/− and wild-type cells, phosphoproteomic analysis was conducted. A total of 21, 37, and 781 phosphopeptides on 21, 37, and 499 proteins were identified in control (N/N), single LPS stimulation (N/100), and sequential LPS induction (100/100) samples, respectively (Figure 2A) (Appendix A). Interestingly, of the 781 differentially quantified phosphoproteins, 381 phosphoproteins, including protein kinase C-β type II (PRKCB), were significantly down-regulated in LPS-tolerant FcGRIIb−/− macrophages, as illustrated by the volcano plot analysis (Figure 2B). Using the DAVID Functional Annotation Clustering Tool, the functional annotation clustering identified PRKCB mediated signaling downstream of a phagocytosis pathway (Figure 2C) and other clusters are listed in Appendix A. Indeed, the downregulation of phosphoproteins in the FcGR-mediated phagocytosis pathway might be associated with the induction of LPS-tolerance in FcGRIIb−/− macrophages. Then, the 122 differently expressed proteins were subjected to Gene ontology (GO) analysis using the PANTHER tool. The functional classifications and the number percentages of the proteins we found are shown in Figure 2D. Among several phosphoproteins in this group, the serine-threonine protein kinase AKT1 (Akt1), sphingosine kinase (SPHK), P21-Activated Kinase 1 (PAK1), and protein kinase C-β type II (PRKCB) (Figure 2D) are important mediators for phagocytosis, as determined by KEGG pathway analysis. While there are numerous studies of Akt signaling in macrophages, focusing on several cell activities [15], some studies mention protein kinase C as the association mediator of TLR signaling [16,17]. Thus, the defect of a specific isoform of protein kinase C (-β type II) might be responsible for the extensive LPS tolerance in FcGRIIb−/− macrophages. To explore whether PRKCB could enhance macrophage functions, PRKCB was over-expressed in RAW264.7 cells (Figure 3A). Indeed, the over-expression of PRKCB enhanced phagocytic activity and cytokine production in both a single LPS stimulation and LPS tolerance (Figure 3B,C). Therefore, PRKCB over-expression could prevent the reduced phagocytic activity and decreased cytokine production in LPS tolerance in a wild-type monocyte/macrophage cell line (RAW264.7).

### 2.3. Protein Kinase C Inducer, PMA, Increased Protein Kinase C-β Type II (PRKCB) and Attenuated LPS Tolerance In Vitro and In Vivo 

To determine the influence of protein kinase C on LPS stimulation in macrophages, PMA, a well-known protein kinase C activator, was used [18]. With a single LPS stimulation (N/100), PMA enhanced cytokine production in wild-type cells, but not FcGRIIb−/− (Figure 4A–C). Despite the similarly increased PRKCB in both strains of macrophages (Figure 4G), cytokine production in FcGRIIb−/− was higher than in the wild-type specimens. On the other hand, in LPS tolerance (100/100), PMA increased cytokine production in both wild-type and FcGRIIb−/− macrophages in at least one time-point of the incubation (Figure 4D–F) and also induced PRKCB expression in both strains (Figure 4H). Hence, PMA only enhanced cytokine production in wild-type macrophages in a single LPS stimulation, but attenuated LPS-tolerance in both wild-type and FcGRIIb−/− cells. Therefore, PMA might be beneficial for the attenuation of LPS tolerance in vivo. We tested PMA in a mouse model of LPS tolerance in both wild-type and FcGRIIb−/− mice. Indeed, lower serum IL-6 in FcGRIIb−/− mice compared with wild-type specimens after LPS-tolerance induction was found and PMA increased serum cytokines in both FcGRIIb−/− and wild-type cells with LPS- tolerance (Figure 5A–C). However, PMA increased PRKCB in spleens of FcGRIIb−/− mice, but not in those of wild-type mice (Figure 5D).

The inflammatory responses are important for disease control, especially in the early phase of sepsis and LPS tolerance-induced macrophage dysfunction decreases sepsis severity [10]. Protein kinase C activation in a mouse model of polymicrobial sepsis after LPS tolerance (CLP with LPS pre-conditioning; see Section 4) was performed as a proof of concept of sepsis attenuation in the immune exhaustion phase. 

Indeed, PMA administration induced spleen *PRKCB* at 6 h after CLP and attenuated sepsis severity, as determined by survival, renal injury, liver injury, bacterial burdens, and increased serum cytokine levels in FcGRIIb−/− mice (Figure 6). In wild-type mice, PMA also increased spleen PRKCB levels (Figure 6D), which was possibly responsible for enhanced serum proinflammatory cytokines (TNF-α and IL-6) (Figure 6F,G), along with reduced blood bacterial burdens at 6h, but not at 24 h, post-CLP (Figure 6E). Of note, LPS-tolerance of FcGRIIb−/− mice was more severe than for wild-type mice, as demonstrated by the lower serum cytokines in FcGRIIb−/− at 6h post-CLP (after LPS tolerance) compared with wild-type mice (Figure 6 F–H, left side). PMA attenuated sepsis (after LPS tolerance) in FcGRIIb−/− mice, but not in wild-type mice, possibly due to the difference in the severity of LPS tolerance between mouse strains. 

## 3. Discussion

Because of the high incidence of FcGRIIb dysfunction polymorphism in Asian populations [2], FcGRIIb−/− mice might be one of the good representative models of lupus in Asian populations. The activity of the endotoxin against FcGRIIb−/− macrophages supported the idea of cross-talk between Fc gamma receptors and Toll-like receptor-4, as previously described [19]. 

### 3.1. The Characteristics of Prominent Endotoxin Exhaustion of FcGRIIb−/− Macrophage

Although reduced cytokine production after the second LPS stimulation is a main characteristic of macrophage endotoxin-tolerance [20], the data on other macrophage functions after LPS tolerance induction is still lacking. Indeed, in wild-type macrophages, LPS tolerance reduced cytokine production without the influence on phagocytosis and microbicidal activity. In contrast, LPS tolerance in FcGRIIb−/− macrophages reduced all these functions, at least in part, supporting the notion of increased susceptibility to infection in FcGRIIb−/− mice over the wild-type mice, as we reported previously [10]. While LPS tolerance affected only TNF-α and IL-6 production in wild-type macrophages, LPS tolerance also impaired other pathogen control mechanisms (phagocytosis and microbicidal activity) in FcGRIIb−/− cells as a mechanism possibly responsible for the significant LPS exhaustion in lupus. Indeed, the importance of LPS tolerance in bacterial sepsis has been described [20] and infection with Gram negative bacteria (the source of LPS) is one of the leading causes of death in lupus [21]. Likewise, enhanced endotoxin tolerance might be, at least in part, responsible for the increased mortality rate of patients with lupus, which is a major cause of death of young females in the US [22]. Hence, the mechanistic study of LPS tolerance might improve the infection control in patients with lupus. 

### 3.2. Protein Kinase C-β Type II (PRKCB), One of the Responsible Mediators of the Prominent LPS Tolerance in FcGRIIb−/−

Our phosphoproteomic analysis demonstrated a lower abundance of several phosphoproteins in the FcGRIIb−/− macrophages over wild-type ones. Among them, PRKCB was an intriguing protein responsible for several signaling cascades [23]. In addition, PRKCB also decreased in LPS-tolerant FcGRIIb−/− macrophages and in the spleens of FcGRIIb−/− mice with sequential LPS administration. Indeed, several subtypes of protein kinase C are associated with immune responses [24]. The defect of protein kinase C-β type I and -β type II in mice induced a marked immunodeficiency [25]. However, the research involving this specific type of protein kinase C (-β type II: PRKCB) in macrophages is still limited. The KEGG pathway analysis demonstrated that the PRKCB pathway was downstream of FcGR signaling (Figure 2D) and PRKCB overexpression in macrophages enhanced phagocytosis and cytokine production after a single and sequential LPS stimulation. Although only a single cell line was used for the gene overexpression, the data suggest the influence of PRKCB downstream of LPS stimulation in macrophages. 

### 3.3. PMA Is a Potential Regulator of LPS Tolerance Attenuation

PMA is one of the phorbol esters, which are generally known as mutagenesis activity inducers and protein kinase C activators with toxicity in animals [26]. However, PMA at a low dose has been successfully used in mice to enhance protein kinase C activity in vivo [18]. Indeed, PMA induces several members of the protein kinase C family, including PRKCB, with a limited effect on the single LPS stimulated FcGRIIb−/− macrophages (high responses due to the loss of inhibitory signaling). However, PMA attenuated LPS tolerance in both wild-type or FcGRIIb−/− strains, as determined by the increased cytokine levels in both macrophages and mice. PMA attenuated blood bacterial burdens and increased serum cytokines at 6 h in both wild-type and FcGRIIb−/− mice in a model of sepsis immune exhaustion (CLP after LPS tolerance), but only improved survival in FcGRIIb−/−. As a proof of concept, the induction of protein kinase C may be examined for attenuation of the immune-exhaustion phase in sepsis and protein kinase C-β type II might be an important mediator responsible for the prominent LPS tolerance in lupus. As such, the use of several biomarkers (immune exhaustion detection and PMA toxicity monitoring) might be an interesting strategy. 

In conclusion, the prominent LPS-tolerance of FcGRIIb−/− macrophages over wild-type ones was, in part, due to the significantly depleted PRKCB. The levels of PRKCB were reduced in LPS-tolerant macrophages and in the spleens of LPS-tolerant mice. PMA attenuated LPS tolerance and reduced the consequent severity of sepsis in the LPS tolerance model, predominantly in FcGRIIb−/− mice over wild-type mice. PRKCB might be considered a novel target for the treatment of LPS-tolerance in lupus. 

## 4. Materials and methods

### 4.1. Animals

FcGRIIb−/− mice on a C57BL/6 background were obtained from Dr. Silvia Bolland (NIAID, NIH, MD, USA) and wild-type female C57BL/6 mice were purchased from the National Laboratory Animal Center in Nakhon Pathom Province, Thailand. Eight-week-old female mice were used following the procedures approved by the Faculty of Medicine, Chulalongkorn University in accordance with the National Institutes of Health (NIH) criteria (SST 002/2559, May 2016). The representative result of mouse genotyping is shown in Appendix A.

### 4.2. Preparation of Bone Marrow-Derived Macrophages and In Vitro LPS Stimulations

Bone marrow-derived macrophages (BMM) were prepared following an established procedure [27]. Briefly, BM cells from mouse femurs were obtained and cultured for seven days in Dulbecco’s Modified Eagle Medium (DMEM, Sigma-Aldrich, St. Louis, MO, USA) supplemented with L929-conditioned media in a humidified incubator at 37 °C with 5% CO_2_. The harvested cells were confirmed for the macrophage phenotype with anti-F4/80 and anti-CD11c antibody staining (Bio Legend, San Diego, CA, USA) by flow cytometry. 

LPS tolerance was induced in macrophages by the addition of 100 ng/ml LPS diluted in 100 µl/well with 1 × 10^5^ cells/well seeded in a 96-well plate following a previously described protocol [10]. Macrophage stimulation protocol for phosphoproteomic analysis included (i) single LPS stimulation (N/100), where culture media without LPS was used in the first 24 h, followed by washing and re-stimulation with LPS-containing media; (ii) LPS tolerance (sequential LPS stimulation; 100/100), where LPS-containing media was used for the first 24 h, followed by washing and re-stimulation with LPS-containing media; and (iii) the control group (N/N), where only culture media without LPS were used before and after washing. In addition, N/100 and 100/100 protocols were used to test the influence of phorbol 12-myristate 13-acetate (PMA), a protein kinase activator, on macrophage functions. Then, PMA (Sigma-Aldrich, St. Louis, MO, USA) at 100 ng/ml, or culture media control, was added 10 min before LPS induction in N/100 and before the second LPS stimulation in the 100/100 protocol. 

### 4.3. Macrophage Functions (Cytokine Production, Phagocytosis, and Microbicidal Activity)

Several macrophage functions were tested after the single and sequential LPS stimulation. Cytokines in the supernatants were analyzed by ELISA (Thermo-Scientific, Rockford, IL, USA). A phagocytic activity assay was performed following a previous study [28]. Briefly, 200 µg/ml of zymosan conjugated with 40 kDa fluorescein isothiocyanate dextran (FITC-dextran) (Sigma-Aldrich, St. Louis, MO, USA) was added to the cells, which were then incubated for 1 h at 37 °C in 5% CO_2_. Following this, the extracellular fluorescence and non-ingested FITC-dextran were quenched and removed by adding Trypan Blue and vigorous washing with phosphate buffer solution (PBS), respectively. The fluorescent positive cells representing FITC-phagocytized cells were analyzed using a fluorescent intensity reader at 492 nm excitation and 518 nm emission wavelengths (Verioskan Flash microplate reader, Thermo-Scientific, Rockford, IL, USA) and an Olympus IX81 inverted fluorescence microscope (Tokyo, Japan). Macrophage microbicidal activity was also measured, as per previously described protocol [10], with the incubation of a 1 × 10^7^ colony forming unit (CFU) of *E. coli* (American Type Culture Collection, ATCC, Manassas, VA, USA) in BMM at 1 × 10^5^ cell/ well (96-well plate) using 25 µL of normal mouse serum as an opsonin. Then, the non-phagocytized bacteria were removed by washing and incubated further for 1 h with 100 µL of gentamicin 100 µg/mL to eradicate viable extracellular bacteria. Subsequently, the plate was washed and induced cell lysis by 200 µL of sterile water/well, before being plated on Trypticase soy agar in the serial dilution. Bacterial count was evaluated after 24 h of 37 °C incubation. 

### 4.4. Dimethyl Labeling, Phosphopeptide Enrichment, and Fractionation

The cell pellet of macrophages from four mice of each group was lysed with 5 % deoxycholic acid sodium salt (SDC) plus 1× phosphatase inhibitor cocktails (Thermo-Scientific Rockford, IL, USA). Protein concentration was determined using a bicinchoninic acid (BCA) assay (Bio-Rad laboratories, Hercules, CA, USA), and 300 µg protein from each group was mixed with 10 mM of dithiothreitol (DTT) at 56 °C for 30 min and alkylated by 40 mM iodoacetamide (IAA) at room temperature for 30 min (excess IAA was also quenched). Then, the preparation was diluted to 1% (*v*/*v*) in 100 mM triethylammonium bicarbonate (TEAB) followed by trypsin at an enzyme to substrate ratio of 1:100. After that, the peptide solution was lyophilized by vacuum centrifugation and dissolved in 100 mM TEAB (100 µL) and determined by a peptide quantification assay (Thermo-Scientific Rockford, IL, USA). Peptides were quantitatively compared using stable isotopic dimethyl labeling with light labeling (formaldehyde and sodium cyanoborohydride) in the wild-type group and intermediate labeling (deuterated formaldehyde and sodium cyanoborodeuteride) in the FcGRIIb−/− group. The peptide solution was mixed with 25% ammonia (30 µL) to neutralize excess formaldehyde, and then mixed with 100% formic acid (15 µL) to acidify the solution. For TiO_2_ StageTips, a layer of C8 disks (Empore, Sigma) was put into 200 µL tips to perform TiO_2_ enrichment and TiO_2_ beads (GL Sciences Inc., Mainz, Germany) were preconditioned with 500 µL of binding buffer [29]. Then, the peptides were dissolved in 200 µL of binding buffer with 500 µL of equilibrated bead slurry. The mixture was transferred to stage tips and spun at 1000× *g*, 5 min. Subsequently, the beads were washed two times with 80% (*v*/*v*) acetonitrile, 5 % (*v*/*v*) trifluoroacetic acid (TFA), and the bound-peptides were eluted with 180 µL of 10% NH_4_OH, pH 10.5. The supernatant was dried and stored at −80 °C. Next, the peptides were separated into 10 fractions to reduce complexity using the Pierce High pH Reversed-Phase Peptide Fractionation Kit (Thermo-Scientific Rockford, IL, USA). Finally, the elutes from each fraction were dried in a SpeedVac centrifuge before LC-MS/MS analysis.

### 4.5. LC-MS/MS and Analysis

The peptides were submitted to LC-MS/MS by a Q-Exactive Plus mass spectrometer (Thermo-Scientific Rockford, IL, USA). The MS methods included a full MS scan at a resolution of 70,000, followed by 10 data-dependent MS2 scans at a resolution of 17,500. The normalized collision energy of HCD fragmentation was set at 30%. An MS scan range of 400 to 1600 *m*/*z* was selected and precursor ions with unassigned charge states, a charge state of +1, or a charge state of greater than +8, were excluded. A dynamic exclusion of 30 s was used. Using Maxquant Software (version 1.5.1.2), the MS raw data files were searched against a composite database containing the forward and reversed peptide sequences of the Mouse Uniprot Database. The search parameters were set for the following fixed modifications: carbamidomethylation of cysteine (+57.02146 Da), as well as light and medium dimethylation of N-termini and lysine (+28.031300 and +32.056407 Da). For variable modification, the oxidation of methionine (+15.99491 Da) and phosphorylation of serine, threonine, and tyrosine were set. A maximum of four modifications and two missed cleavages per peptide were allowed. Parent and fragment mono-isotopic mass errors were set at 10 and 0.2 ppm, respectively. A target–decoy approach was used to limit the false discovery rate of the identified peptides to less than 1%. The wild-type mice were used as denominators to generate ratios of (FcGRIIb−/−)/(wild-type) mice. Significantly differentially regulated-proteins were determined by unpaired *t*-tests with a *p*-value < 0.05. In addition, Gene ontology analysis (GO) and functional enrichment of the significance of proteins in the molecular function, biological process, and cellular component categories were performed with (http://david.abcc.ncifcrf.gov/) [30] and the Protein Analysis through Evolutionary Relationships (PANTHER) classification system (http://www.pantherdb.org) [31]. The Kyoto encyclopedia of genes and genomes (KEGG) PATHWAY (http://www.genome.jp/kegg) [32] was used to perform signaling pathway enrichment analysis with *p* < 0.05 as a cut-off criterion.

### 4.6. Protein Kinase C-β Type II (PRKCB) Overexpression In Vitro 

Because protein kinase C-β type II (PRKCB) (Thr641; the insertion of phosphate group onto threonine at 641 position of amino acid sequence) was one of the interesting proteins from the phosphoproteomic analysis, the overexpression of this gene in RAW264.7 cells (ATCC, Manassas, VA, USA) was performed to explore its importance in macrophage function. The plasmid construct with a mouse *Prkcb* gene cDNA expression clone fused with an REP plasmid (Sino-biological Inc, Wayne, PA, USA) was transfected into the cells. Plasmid DNA constructs were transformed into high efficiency DH5α competent cells (NEB) for propagation, and purified by the GeneJET plasmid miniprep kit (Thermo-Scientific Rockford, IL, USA), following the manufacturer’s instructions. For cell transfection, RAW264.7 cells were transfected with PRKCB-REP, an empty REP vector using TurboFect transfection reagent (Thermo-Scientific Rockford, IL, USA), according to the manufacturer’s instructions, with 6 h incubation for plasmid expression. The transfected RAW264.7 cells were incubated in fetal bovine serum (FBS)-DMEM medium with a single dose or sequential doses of LPS (100 ng/mL) and were further tested for macrophage functions as mentioned above. 

### 4.7. Western Blot Analysis

BMM or RAW264.7 cells were pelleted and washed with 1× PBS. Once cells were pelleted, the supernatants were removed. The cells were re-suspended and lysed in RIPA lysis buffer, 1× Protease inhibitors, and 1× Phosphatase inhibitors (Thermo-Scientific Rockford, IL, USA), and spleens were homogenized for PRKCB detection. The cell mixture or spleen preparations were sonicated on ice and then centrifuged at 1500× *g* at 4 °C and the protein quantification was performed by the BCA assay (Pierce BCA Protein Assay, Thermo Scientific). The protein (20 µg) was separated in 10% sodium dodecyl sulfate (SDS) polyacrylamide gel, transferred to a nitrocellulose membrane, detected by Phospho-PRKCB (Thr641) primary antibodies (Thermo-Scientific Rockford, IL, USA), probed with a rabbit anti-mouse IgG horseradish peroxidase (HRP)-conjugated secondary antibody (Santa Cruz Biotechnology), and detected by an enhanced chemiluminescence rapid step chemiluminescence detection system (Thermo-Scientific Rockford, IL, USA). The rabbit monoclonal antibodies of glyceraldehyde 3-phosphate dehydrogenase (GAPDH) and beta-actin (Cell Signaling) were used as the housekeeping genes for the preparations of cells and spleens, respectively. 

### 4.8. Endotoxin-Tolerance Mouse Model 

To test endotoxin-tolerance in vivo, FcGRIIb−/− and wild-type mice were injected intra-peritoneally (IP) with three separate doses of endotoxin (LPS of *Escherichia coli* 026:B6, Sigma-Aldrich, St. Louis, MO, USA) diluted in PBS at 0.8 mg/kg and then 4 mg/kg three and six days later. Phorbol 12-myristate 13-acetate (PMA) (Sigma-Aldrich) at 0.2 mg/kg was administered 10 min before the second and third doses of LPS to activate protein kinase C in mice (modified from [18]). In parallel, PBS was administered in the control group. Blood was collected through tail vein nicking to measure the time-course of serum cytokines (TNF-α, IL-6, and IL-10) by the ELISA assay (Thermo-Scientific Rockford, IL, USA) and spleens were collected for Western blot analysis at sacrifice by cardiac puncture under isoflurane anesthesia at 24h post-third dose of LPS injection. A total of 24 mice were used for the LPS-tolerance model.

### 4.9. Cecal Ligation and Puncture with Endotoxin Pre-Conditioning Mouse model 

Cecal ligation and puncture (CLP) with an endotoxin pre-conditioning mouse model was performed to explore the influence of immune exhaustion upon polymicrobial infection. Then, the immune exhaustion was induced by two separate doses of LPS administration before CLP surgery. LPS at 0.8 mg/kg followed by 4 mg/kg three days later was administered for endotoxin tolerance induction in FcGRIIb−/− and wild-type mice and CLP surgery was induced 12 h after the second dose of LPS [10]. In addition, two doses of PMA (Sigma-Aldrich) at 0.2 mg/kg/dose (as protein kinase C activation) or PBS (as control) was administered 10 min before the second LPS injection and before CLP surgery, respectively. Of note, CLP procedures by ligation at 10 mm from the cecal tip and double puncture with a 21-gauge needle under isoflurane anesthesia were performed as previously described [10,18]. Blood was collected at the time of sacrifice through cardiac puncture under isoflurane anesthesia for (i) blood bacterial burdens, determined by bacterial colony enumeration in blood agar (Oxoid, Hampshire, UK) after 24 h incubation at 37 °C; (ii) serum creatinine (Scr) (QuantiChrom Creatinine Assay, DICT-500, BioAssay, Hayward, CA, USA) and alanine transaminase (ALT) (EnzyChrom ALT assay, EALT-100, BioAssay) for kidney and liver injury, respectively; and iii) serum cytokines by ELISA assays (Thermo-Scientific). Spleens were collected to determine PRKCB levels by Western blot analysis. Therefore, 60 mice were used for the survival study, together with 48 and 50 mice for 6 h and 24 h time-points, respectively.

### 4.10. Statistical Analysis 

The data were shown as the mean ± standard error (SE) and the differences between groups were examined by the unpaired student *t*-test or one-way analysis of variance (ANOVA) with Tukey’s comparison for the analysis of experiments with two or three groups, respectively. *p* values < 0.05 are considered statistically significant. SPSS 11.5 software (SPSS Inc., Chicago, IL, USA) was used for all statistical analysis.

## Figures and Tables

**Figure 1 ijms-20-01354-f001:**
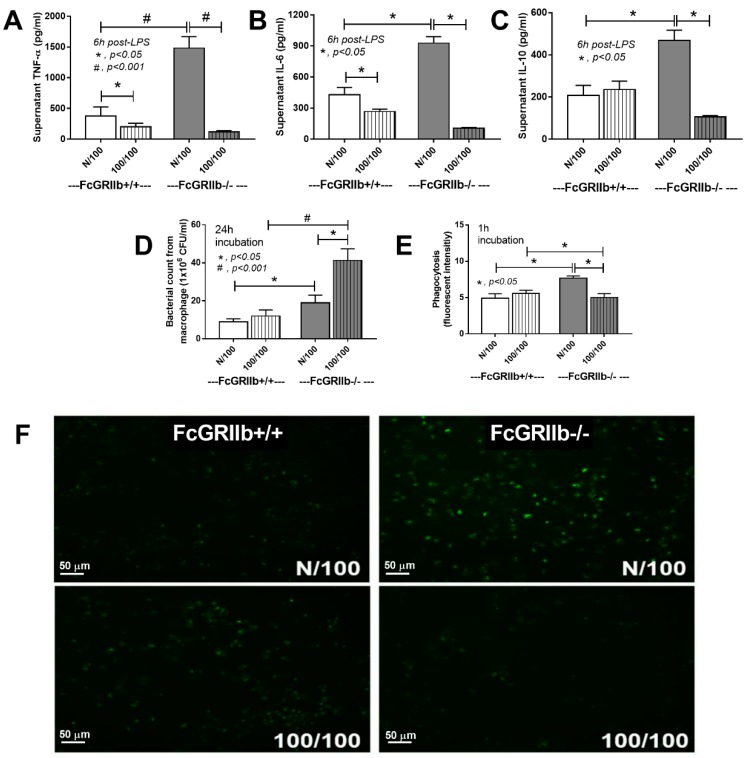
The characteristics of FcGRIIb+/+ (WT) versus FcGRIIb−/− macrophages (KO) after a single LPS stimulation (N/100) and sequential LPS activation (100/100; LPS tolerance) shown as cytokine activation (**A**–**C**), microbicidal activity (**D**), and phagocytosis with mean fluorescent intensity (**E**) and the representative of phagocytosis by a fluorescent microscope (green dots were phagocytosed FITC-dextran conjugated zymosan) (**F**). Independent experiments were performed in triplicate.

**Figure 2 ijms-20-01354-f002:**
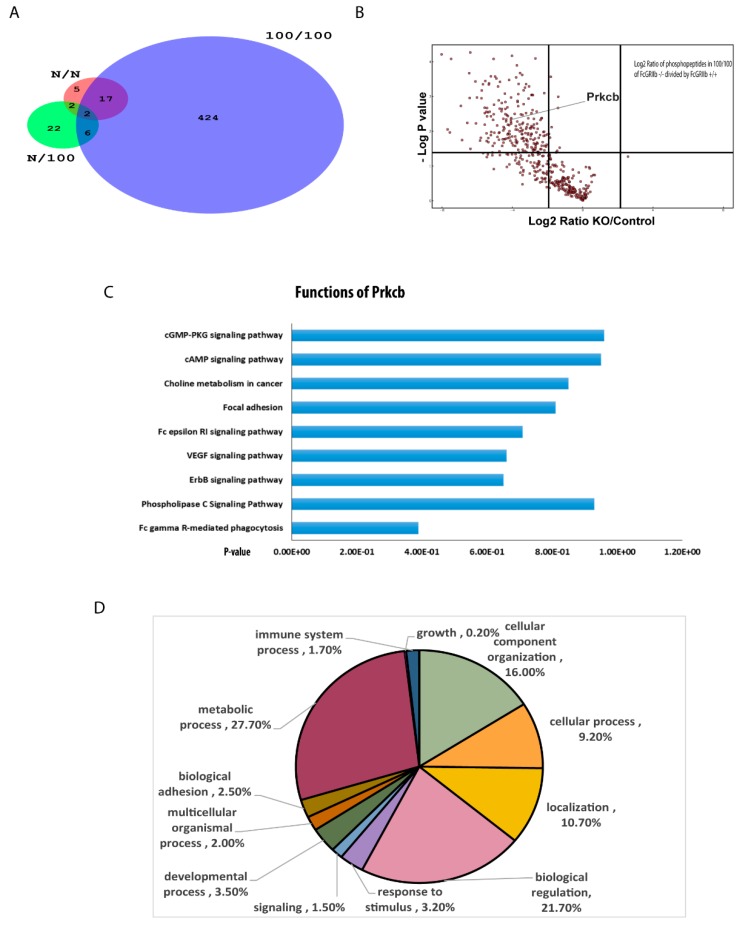
(**A**) Venn diagram demonstrating the different number of phosphoproteins from FcGRIIb+/+ macrophages in comparison with FcGRIIb−/− after a single LPS stimulation (N/100), sequential LPS activation (100/100; LPS-tolerance), and the control (N/N) (*n* = 4/each group); (**B**) the volcano plot analysis of downregulated phosphoproteins from the sequential LPS activation of FcGRIIb−/− compared with FcGRIIb+/+ macrophages; (**C**) pathway analysis clusters (DAVID) of the significantly altered phosphopeptides in FcGRIIb+/+ compared with FcGRIIb−/− in LPS-tolerance (100/100); (**D**) GO annotation of differentially expressed proteins (FcGRIIb+/+ compared with FcGRIIb−/−) in biological processes; and (**E**) the enrich pathway of the phosphoproteome of FcGRIIb+/+ compared with FcGRIIb−/− was related to the phagocytosis pathway that MPHIm, Akt, PKC, SPHK, PAK1, Vav, and DOCK180 (in black star) were the proteins involved in this pathway.

**Figure 3 ijms-20-01354-f003:**
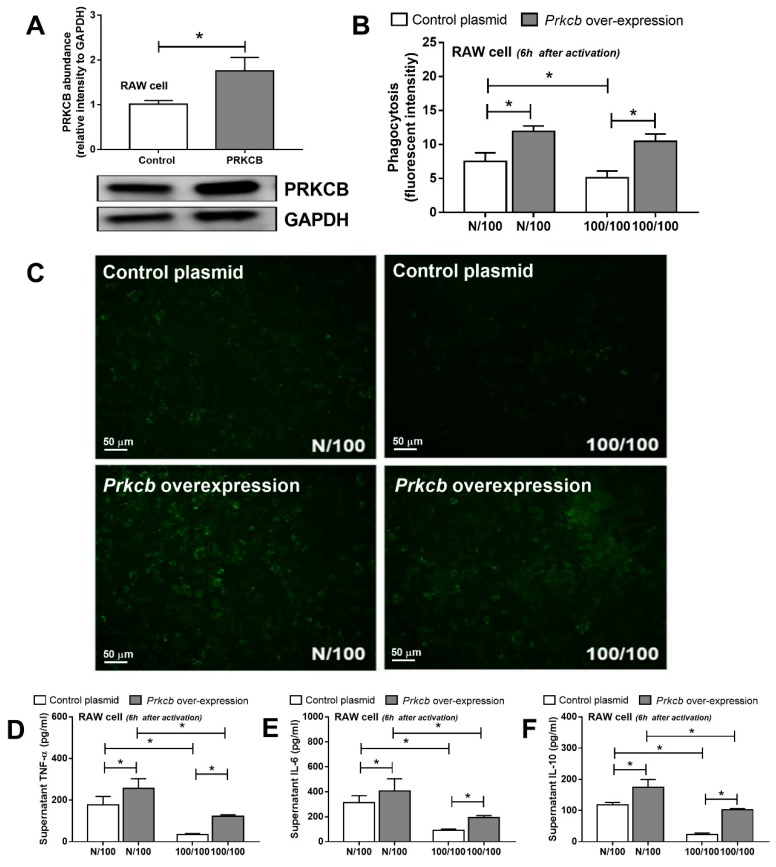
The abundance of protein kinase C-β type II (PRKCB) shown by Western blot analysis from a monocyte/macrophage cell line (RAW264.7) with and without PRKCB overexpression (**A**) and macrophage functions after a single LPS stimulation (N/100) and sequential LPS activation (100/100; LPS-tolerance), as determined by the phagocytosis assay, (**B**) with representative images (green dots were phagocytosed FITC-dextran conjugated zymosan) (**C**) and cytokines production (**D**–**F**). Individual experiments were done in triplicate. * *p* < 0.05.

**Figure 4 ijms-20-01354-f004:**
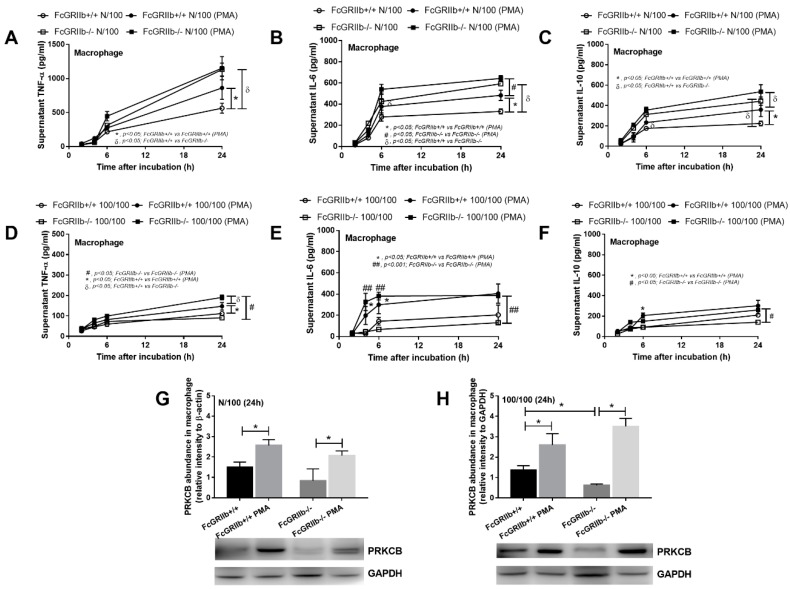
Cytokine levels in supernatants from FcGRIIb+/+ and FcGRIIb−/− macrophages after a single LPS stimulation (N/100) (**A**–**C**), sequential LPS activation (100/100; LPS tolerance) (**D**–**F**), and the abundance of protein kinase C-β type II (Prkcb) (**G**,**H**) with and without PMA activation. Individual experiments were done in triplicate.

**Figure 5 ijms-20-01354-f005:**
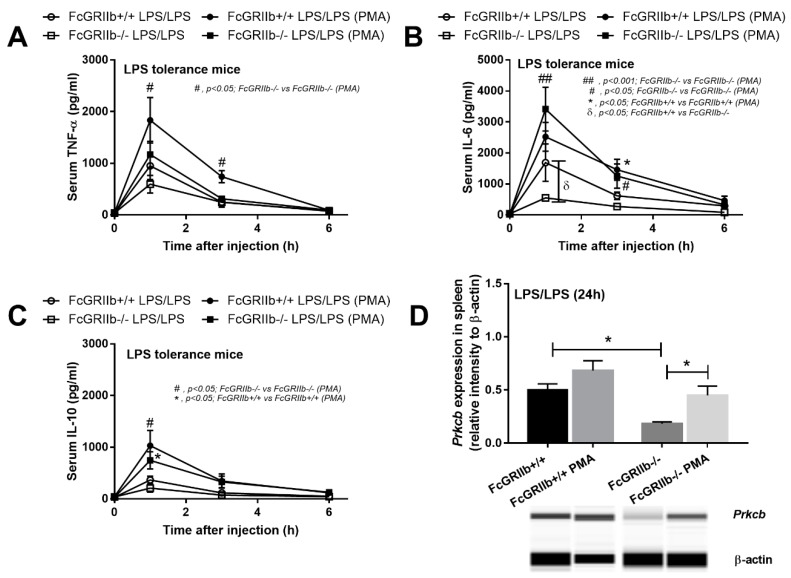
Serum cytokines from FcGRIIb+/+ and FcGRIIb−/− mice after sequential LPS activation (LPS-tolerance by three doses of LPS; see methods) (LPS/LPS) (**A**–**C**) and the expression of protein kinase C-β type II (*Prkcb*) in the spleen by qRT-PCR (**D**) with and without PMA activation (*n* = 5–6/time-point for **A**–**C** and *n* = 4/ group for **D**).

**Figure 6 ijms-20-01354-f006:**
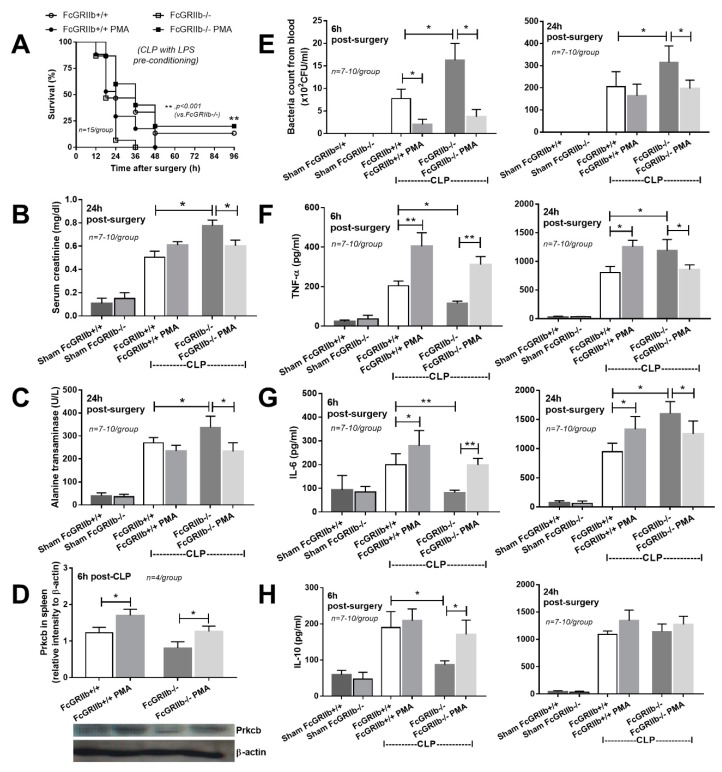
Sepsis severity of cecal ligation and puncture (CLP) after LPS preconditioning (see Section 4) as determined by survival analysis (**A**), renal injury (serum creatinine) (**B**), liver injury (serum alanine transaminase) (**C**), the abundance of protein kinase C-β type II (PRKCB) in the spleen (**D**), blood bacterial count (**E**), and serum cytokines (**F**–**H**) (6 and 24 h for **E**–**H**) with and without PMA administration (*n* = 15/ group for A, *n* = 4/ group for **D** and *n* = 7–10/group for others).

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
