# Peer review of "Decreased Protein Kinase C-β Type II Associated with the Prominent Endotoxin Exhaustion in the Macrophage of FcGRIIb−/− Lupus Prone Mice is Revealed by Phosphoproteomic Analysis"

_ijms, 2019, doi:10.3390/ijms20061354_

Reviewer 1 Report

In the present article, Ondee et al treated lupus-prone mice with a defect in macrophage function (FcGRIIb defective) with LPS. They showed that, after a single LPS treatment, an increase in pro-inflammatory cytokines was observed. However, after having induced LPS tolerance, the ability to produce such cytokines was markedly reduced abd eve macrophage phagocytosis was suppressed. Additionally, they demonstrated that this phenomenon could be mediated by Prkcb, since its overexpression/stimulation may reversed the reduced phagocytic activity and decreased cytokine production.

The rationale of CLP experiment is quite foggy. Indeed LPS is often generated by gut microbiota. In this case LPS was administered by endo-peritoneal injection and no intestinal sepsis or gut microbiota manipulation was performed. It would have been more interesting to give LPS via enema or oral route and then observe the effects. If Authors meant to induce a variation of bacterial charge after cecal ligation, this cannot be demonstrated since bacterial charge was not measured before and after the procedure.

Author Response

Comments and Suggestions for Authors 1

In the present article, Ondee et al treated lupus-prone mice with a defect in macrophage function (FcGRIIb defective) with LPS. They showed that, after a single LPS treatment, an increase in pro-inflammatory cytokines was observed. However, after having induced LPS tolerance, the ability to produce such cytokines was markedly reduced abd eve macrophage phagocytosis was suppressed. Additionally, they demonstrated that this phenomenon could be mediated by Prkcb, since its overexpression/stimulation may reversed the reduced phagocytic activity and decreased cytokine production.

The rationale of CLP experiment is quite foggy. Indeed LPS is often generated by gut microbiota. In this case LPS was administered by endo-peritoneal injection and no intestinal sepsis or gut microbiota manipulation was performed. It would have been more interesting to give LPS via enema or oral route and then observe the effects. If Authors meant to induce a variation of bacterial charge after cecal ligation, this cannot be demonstrated since bacterial charge was not measured before and after the procedure.

ANS: We thank the reviewer for the comment and apologize for the unclear presentation in the previous version of the manuscript. The CLP with LPS pre-conditioning model is a representative model of an infection in immune exhausted situation. Thus the immune activation aim to induce immune response systemically and immune exhaustion was induced by 2 separate doses of LPS. Although the local immune activation by LPS through oral or enema administration is also interesting (this might affect gut-microbiota), the current experiment aim to explore the defect of systemic immune responses. Hence, we mentioned the rational of this model in the method section of the new version manuscript.

 Reviewer 2 Report

Systemic lupus erythematosus is a severe health burden and one key player identified in this disease is FcGRIIb. In this study, Ondee et al. examined the consequences of FcGRIIb deletion on endotoxin tolerance in vivo. The authors have used a comprehensive approach by performing phosphoproteomic analyses and by characterizing bone marrow-derived macrophages as well as macrophage cell lines. However, the impact of this study is clearly limited due to highly observational nature of the experiments as well as by the poor presentation of data. Specific concerns are detailed below.

The authors use primary material from FcGRIIb-/- mice, however, the knockout efficiency has not been demonstrated. Please provide information on the knockout efficiency on mRNA and protein level.

A thorough characterization of FcGRIIb-/- mice is missing. What are the serum parameters of these animals if not triggered with LPS and/or CLP?

This study uses only a single cell line (RAW264.7) to investigate the functions of Prkcb. To avoid showing cell line-specific effects, a second cell line should be used.

It is not always clear, how many mice were used for the respective experiments. Please mention the n-numbers of in the Materials and Methods section as well as in the figure legends.

The authors did not cite the references of the DAVID and KEGG database. Following information is given on the DAVID website: “DAVID users may publish or otherwise publicly disclose the results of using DAVID. Please acknowledge DAVID in your publications by citing the following two references: Huang DW, Sherman BT, Lempicki RA. Systematic and integrative analysis of large gene lists using DAVID Bioinformatics Resources. Nature Protoc. 2009;4(1):44-57. Huang DW, Sherman BT, Lempicki RA. Bioinformatics enrichment tools: paths toward the comprehensive functional analysis of large gene lists. Nucleic Acids Res. 2009;37(1):1-13.” (https://david.ncifcrf.gov/content.jsp?file=citation.htm).
Similarly, on the KEGG website following information is given: Please cite the following article(s) when using KEGG. Kanehisa, M., Sato, Y., Furumichi, M., Morishima, K., and Tanabe, M.; New approach for understanding genome variations in KEGG. Nucleic Acids Res. 47, D590-D595 (2019). Kanehisa, Furumichi, M., Tanabe, M., Sato, Y., and Morishima, K.; KEGG: new perspectives on genomes, pathways, diseases and drugs. Nucleic Acids Res. 45, D353-D361 (2017). Kanehisa, M. and Goto, S.; KEGG: Kyoto Encyclopedia of Genes and Genomes. Nucleic Acids Res. 28, 27-30 (2000). (https://www.genome.jp/kegg/kegg1.html)

In Fig. 4A-F, the x-axes are labeled as “Time after incubation (min)” with a maximum of 24 min. Should this be hours instead of minutes?

PAK1 is the P21-Activated Kinase 1 and not the “p21 protein” as described in line 249.

Centrifugation speed should be given in xg, not in rpm (line 176).

The labeling (probably “p-value”) of the x-axis in Fig. 2C is missing.

There are numerous typing errors and inconsistencies present throughout the entire manuscript suggesting that the manuscript has not been prepared carefully. Besides typing errors (examples: “Laboritory” (line 13), “wild-type cell sand” (line 224)), there are words missing (examples: “in the FcGRIIb-/-.” (line 125/126), “in FcGRIIb-/-.” (line 309 and 365)) as well as double/missing spaces (examples in line 352 and 208). Inconsistencies include “over-expression” (Fig. 3B) or “overexpression” (Fig. 3C) or the symbol for degree Celsius (lines 100 and 113).

Please avoid the term “killing activity” when describing the microbicidal activity of macrophages.

The scale of the y-axes of several graphs should be adjusted (example: Fig. 1 B and C).

The authors should avoid the term “expression” when describing protein amounts. In general, genes are expressed, not proteins. For instance, in Fig. 3A the y-axis of the graph is labeled with “Prkcb expression”, suggesting that mRNA levels were detected using qRT-PCR. In fact, the authors show a quantification of the bands detected after Western blot analysis. Besides the fact that “Prkcb” should be capitalized when describing protein levels, the term “expression” could lead to confusion. In addition, Fig. 5D looks highly similar to Fig. 3A with one difference: in Fig. 5D indeed mRNA was analyzed and not protein (even though “mRNA” or “qRT-PCR” were not mentioned in the legends to make that clear.

As mentioned above, the figure legends should be more detailed. They should include the methods/databases used, n-numbers and briefly describe the meaning/consequence of the results. For instance, in the legend of Fig. 2E the authors did not provide the information that this image was obtained after performing KEGG pathway analysis, what the results mean and KEGG was not cited in the manuscript. Besides that, the font in Fig. 2E is too small and the pink stars are hard to see.

Please provide information on the phosphatase inhibitor cocktails that were used (line 116).

To help the reader understanding the purpose of the experiments, please start each result section by giving a short introduction (“To test whether […], we performed […]”).

The PANTHER tool (line 247) should also be mentioned in the Materials and Methods section.

The authors did not pay attention to the nomenclature. Human gene names should be written in capital letters and be italicized (PRKCB). Human proteins are written in capital letters (PRKCB). Murine gene names should be written in small letters (only the first letter is capitalized) and be italicized (Prkcb). Murine proteins are written in capital letters (PRKCB).

Please define meanings of acronyms the first time they are being used.

Author Response

Comments and Suggestions for Authors 2

Systemic lupus erythematosus is a severe health burden and one key player identified in this disease is FcGRIIb. In this study, Ondee et al. examined the consequences of FcGRIIb deletion on endotoxin tolerance in vivo. The authors have used a comprehensive approach by performing phosphoproteomic analyses and by characterizing bone marrow-derived macrophages as well as macrophage cell lines. However, the impact of this study is clearly limited due to highly observational nature of the experiments as well as by the poor presentation of data. Specific concerns are detailed below.

 The authors use primary material from FcGRIIb-/- mice, however, the knockout efficiency has not been demonstrated. Please provide information on the knockout efficiency on mRNA and protein level.

ANS: We thank the reviewer for the comment. We provide the genotyping confirmation in the supplementary figure 1 in the new version of the manuscript.

 A thorough characterization of FcGRIIb-/- mice is missing. What are the serum parameters of these animals if not triggered with LPS and/or CLP?

ANS: We added this information in the sham column in the new figure 6. Because asymptomatic lupus mice (FcGRIIb-/- at 8 wk old) were used, then all of the presented baseline values were the same as in WT.

 This study uses only a single cell line (RAW264.7) to investigate the functions of Prkcb. To avoid showing cell line-specific effects, a second cell line should be used.

ANS: We thank the reviewer for the comment and agree that the second cell-line could better highlight the function of Prkcb. However, the association of Prkcb, especially in enhanced phagocytosis and increased cell activity is well-known as we mentioned in the reference number 27 and 28. Moreover, the enhance Prkcb expression by PMA also support Prkcb functions from RAW cell experiments. Hence, we mentioned the utilization of a single cell-line as a limitation of our study in the discussion of the new version manuscript as following; “Although only a single type of cell-line was used for the gene over-expression, this data suggests the influence of Prkcb downstream of LPS stimulation in macrophages. ”

 It is not always clear, how many mice were used for the respective experiments. Please mention the n-numbers of in the Materials and Methods section as well as in the figure legends.

ANS: We thank the reviewer for the comment and added the number of mice in method section and figure legends.

 The authors did not cite the references of the DAVID and KEGG database. Following information is given on the DAVID website: “DAVID users may publish or otherwise publicly disclose the results of using DAVID. Please acknowledge DAVID in your publications by citing the following two references: Huang DW, Sherman BT, Lempicki RA. Systematic and integrative analysis of large gene lists using DAVID Bioinformatics Resources. Nature Protoc. 2009;4(1):44-57. Huang DW, Sherman BT, Lempicki RA. Bioinformatics enrichment tools: paths toward the comprehensive functional analysis of large gene lists. Nucleic Acids Res. 2009;37(1):1-13.” (https://david.ncifcrf.gov/content.jsp?file=citation.htm).
Similarly, on the KEGG website following information is given: Please cite the following article(s) when using KEGG. Kanehisa, M., Sato, Y., Furumichi, M., Morishima, K., and Tanabe, M.; New approach for understanding genome variations in KEGG. Nucleic Acids Res. 47, D590-D595 (2019). Kanehisa, Furumichi, M., Tanabe, M., Sato, Y., and Morishima, K.; KEGG: new perspectives on genomes, pathways, diseases and drugs. Nucleic Acids Res. 45, D353-D361 (2017). Kanehisa, M. and Goto, S.; KEGG: Kyoto Encyclopedia of Genes and Genomes. Nucleic Acids Res. 28, 27-30 (2000). (https://www.genome.jp/kegg/kegg1.html)  DONE

ANS: We thank the reviewer for the comment and added 3 references (No.18-20) in method.

 In Fig. 4A-F, the x-axes are labeled as “Time after incubation (min)” with a maximum of 24 min. Should this be hours instead of minutes?

ANS: We extremely apologized for this mistake and corrected it accordingly.

 PAK1 is the P21-Activated Kinase 1 and not the “p21 protein” as described in line 249.

ANS: We extremely apologized for this mistake and corrected it accordingly.

Centrifugation speed should be given in xg, not in rpm (line 176).   

ANS: We extremely apologized for this mistake and corrected it accordingly.

 The labeling (probably “p-value”) of the x-axis in Fig. 2C is missing.

ANS: We extremely apologized for this mistake and corrected it accordingly.

 There are numerous typing errors and inconsistencies present throughout the entire manuscript suggesting that the manuscript has not been prepared carefully. Besides typing errors (examples: “Laboritory” (line 13), “wild-type cell sand” (line 224)), there are words missing (examples: “in the FcGRIIb-/-.” (line 125/126), “in FcGRIIb-/-.” (line 309 and 365)) as well as double/missing spaces (examples in line 352 and 208). Inconsistencies include “over-expression” (Fig. 3B) or “overexpression” (Fig. 3C) or the symbol for degree Celsius (lines 100 and 113)

ANS: We extremely apologized for this mistake and corrected it accordingly.

Please avoid the term “killing activity” when describing the microbicidal activity of macrophages.

ANS: We agree and corrected it accordingly.

 The scale of the y-axes of several graphs should be adjusted (example: Fig. 1 B and C).

ANS: We thank the reviewer for the comment and corrected the y scale of fig 1 and fig 6.

 The authors should avoid the term “expression” when describing protein amounts. In general, genes are expressed, not proteins. For instance, in Fig. 3A the y-axis of the graph is labeled with “Prkcb expression”, suggesting that mRNA levels were detected using qRT-PCR. In fact, the authors show a quantification of the bands detected after Western blot analysis. Besides the fact that “Prkcb” should be capitalized when describing protein levels, the term “expression” could lead to confusion. In addition, Fig. 5D looks highly similar to Fig. 3A with one difference: in Fig. 5D indeed mRNA was analyzed and not protein (even though “mRNA” or “qRT-PCR” were not mentioned in the legends to make that clear.

ANS: We extremely apologized for our mistake and thank the reviewer for this important note. We correct y axis and labeling of Fig 3, 4, 6 (protein abundance) and correct Fig 5D (qRT-PCR).   

 As mentioned above, the figure legends should be more detailed. They should include the methods/databases used, n-numbers and briefly describe the meaning/consequence of the results. For instance, in the legend of Fig. 2E the authors did not provide the information that this image was obtained after performing KEGG pathway analysis, what the results mean and KEGG was not cited in the manuscript. Besides that, the font in Fig. 2E is too small and the pink stars are hard to see.

ANS: We thank the reviewer and added the number of mice in legend and information of legend of Fig2E accordingly.

 Please provide information on the phosphatase inhibitor cocktails that were used (line 116).

ANS: We thank the reviewer and added this sentence accordingly.

 To help the reader understanding the purpose of the experiments, please start each result section by giving a short introduction (“To test whether […], we performed […]”).

ANS: We thank the reviewer and added this sentence accordingly.

 The PANTHER tool (line 247) should also be mentioned in the Materials and Methods section.

ANS: We thank the reviewer and added in Materials and Methods section accordingly.

 The authors did not pay attention to the nomenclature. Human gene names should be written in capital letters and be italicized (PRKCB). Human proteins are written in capital letters (PRKCB). Murine gene names should be written in small letters (only the first letter is capitalized) and be italicized (Prkcb). Murine proteins are written in capital letters (PRKCB).

ANS: We thank the reviewer and edited as your suggestion accordingly.

 Please define meanings of acronyms the first time they are being used.

ANS: We thank the reviewer and added this sentence accordingly.

Round  2

Reviewer 1 Report

No further comments

Author Response

Thank you for your comment and suggestion.

Reviewer 2 Report

Besides language and style editing, all comments have been addressed by the authors.

Author Response

Thank you for your comment and suggestion. The manuscript was check a language and edited a style already.